# Efficacy evaluation of the S-adenosylhomocysteine hydrolase inhibitor MSD-914 in rhesus macaques (*Macaca Mulatta*) challenged with Ebola virus by the intramuscular route

Sara C. Johnston[1]*, Anthony T. Ginnetti[2], Nancy A. Twenhafel[3], Sarah L. W. Norris[4], Shiying Chen[5], Josh L. Moore[4], Christopher W. Boyce[2], Ondraya M. Frick[4], Dave N. Dyer[4], Donald J. Marsh[5], Stephen C. Stevens[4¤], Walter F. Knapp[2], Kerry L. Berrier[4], Hui Wan[2], Gregory C. Adam[2], Timothy J. Hartingh[2], Heather L. Esham[4], Jimmy O. Fiallos[4], Eugene L. Blue[4], Willie B. Sifford[4], Jonathan D. Latty[4], Harold L. Mills[4], Nazira A. Alli[4], Ashley E. Piper[1], Aimee I. Goodson[6], J. Matthew Meinig[7], David B. Olsen[2], Linda A. Lieberman[8☯], Rekha G. Panchal[9☯]*

**1** Virology Division, United States of America Army Medical Research Institute of Infectious Diseases, Frederick, Maryland, United States of America, **2** Merck and Co., Inc., West Point, Pennsylvania, United States of America, **3** Pathology Division, United States of America Army Medical Research Institute of Infectious Diseases, Frederick, Maryland, United States of America, **4** Veterinary Medicine Division, United States of America Army Medical Research Institute of Infectious Diseases, Frederick, Maryland, United States of America, **5** Merck and Co., Inc., Rahway, New Jersey, United States of America, **6** Research Program Office, United States of America Army Medical Research Institute of Infectious Diseases, Frederick, Maryland, United States of America, **7** Bacteriology Division, United States of America Army Medical Research Institute of Infectious Diseases, Frederick, Maryland, United States of America, **8** Merck and Co., Inc., Cambridge, Massachusetts, United States of America, **9** Molecular Biology Division, United States of America Army Medical Research Institute of Infectious Diseases, Frederick, Maryland, United States of America

☯ These authors contributed equally to this work.
¤ Current affiliation: Division of Veterinary Resources, National Institutes of Health, Bethesda, Maryland, United States of America.
* sara.c.johnston2.civ@health.mil (SCJ); rekha.g.panchal.civ@health.mil (RGP)

## Abstract

Ebola virus (EBOV) causes a severe and often fatal hemorrhagic fever in humans for which effective postexposure countermeasures are lacking. Herein, we describe the evaluation of an S-adenosylhomocysteine hydrolase inhibitor, MSD-914, using mouse and nonhuman primate (NHP) models of lethal EBOV. Mice were completely protected from severe disease and death at doses as low as 0.31 mg/kg/day administered orally. From the pharmacological data and a toxicokinetic study, a predicted protective dose was selected for rhesus macaques (RMs). Surprisingly, orally administered MSD-914 was unable to protect RMs at doses as high as 0.8 mg/kg/day despite providing similar exposure of the drug to the efficacious dose observed in the mouse model.

**Data availability statement:** All relevant data are within the manuscript and its Supporting Information files.

**Funding:** This project was supported by Defense Threat Reduction Agency (DTRA) under project Number CB11160 and CB10642. The funders had no role in study design, data collection and analysis, decision to publish, or preparation of the manuscript.

**Competing interests:** The authors have declared that no competing interests exist.

## Introduction

Ebola virus (EBOV) is a zoonotic filovirus capable of causing severe hemorrhagic disease in humans, with case fatality rates of 40–60% based on overall cases across all outbreaks. Outbreaks are recurring in Central Africa, with the most recent outbreaks occurring in 2022, with one outbreak associated with *Orthoebolavirus sudanense* (Sudan virus, SUDV) (164 cases and 55 deaths in Uganda), and two separate outbreaks associated with *Orthoebolavirus zairense* (EBOV) (64 confirmed or probable cases and 43 deaths in the Democratic Republic of the Congo) (https://www.cdc.gov/ebola/outbreaks/index.html). The EBOV epidemic of 2013–2016, associated with the Makona variant of EBOV, affected 10 countries, and caused over 11,000 deaths [1–3]. This epidemic had a crippling effect on naïve areas of Africa, and was essentially the first time EBOV spread beyond endemic regions. This epidemic and the recurrent outbreaks in endemic regions reinforce the critical need for effective countermeasures, particularly therapeutics to combat active infection or prevent severe disease following a known exposure.

Efficacy studies are performed using nonclinical animal models of EBOV. Small animal models (mice and guinea pigs) are routinely used for screening of candidates to determine the appropriate dose, schedule, and/or route for administration [4–9]. Immunocompetent mice and guinea pigs are resistant to filovirus infection and hence mouse or guinea pig adapted variants that are lethal in their respective models have been generated. Mice infected with the mouse adapted ebola variant closely mimic certain hallmarks of pathogenesis observed in humans. NHPs are subsequently used for proof-of-concept and the pivotal efficacy evaluations [10–14]. Herein, we describe the evaluation of an EBOV countermeasure, MSD-914, from initial testing in mice (BALB/c) through a proof-of-concept study in NHPs (Rhesus macaques).

MSD-914 is a previously reported S-adenosylhomocysteine hydrolase (AHCY) inhibitor [15]. This category of drugs typically interferes with virus spread by promoting the accumulation of *S*-adenosylhomocysteine which interferes with methyl transferases (MTases), and AHCY inhibitors have shown broad-spectrum antiviral activity against numerous RNA viruses, including rhabdoviruses, filoviruses, arenaviruses, paramyxoviruses, reoviruses, and retroviruses [16–19]. In this study, we evaluated the broad-spectrum *in vitro* anti-filovirus activity of MSD-914 against EBOV, Marburg virus, and Sudan ebolavirus and subsequently its efficacy in the mouse and NHP model of EBOV infection.

## Materials and methods

### In vitro antiviral activity assay

Dose response studies were carried out to determine the *in vitro* potency ($EC_{50}$) and cellular cytotoxicity ($CC_{50}$) of MSD-914 using an image-based immunofluorescence assay, as previously described [20]. Briefly, cells (HeLa or HFF or MRC-5) were seeded in 384 well imaging plates. Next day, cells were pretreated with the compound for 2 hours and then infected with optimized multiplicity of infection (MOI) of 0.5 (MRC-5), 0.8 (HeLa), 20 (HFF) for EBOV (Duncan/Makona) or 1 MOI (HeLa) for *Orthomarburgvirus marburgense* (Marburg virus, MARV) (Ci67) or 0.1 MOI (HeLa) for

SUDV (Gulu). The compound was tested in 8-point dose response with a 3-fold step dilution. Each dose was tested in four technical replicates. At the end of the infection times of 48 hours, cells were fixed in 10% formalin and then subjected to immunofluorescence staining using viral antigen specific antibodies. Viral infection was detected using mouse monoclonal antibodies: anti-GP 6D8 (EBOV), anti-GP 9G4 (MARV) and anti-GP 3C10 (SUDV). A goat anti-mouse-Dylight 488 was used as a secondary antibody. The plates were imaged using the Opera Phenix and image analysis was performed using Harmony software. The Hoechst and CellMask dyes were used to stain nuclei and cytoplasm, respectively. The assay quality of each plate was assessed using the Z′ (Z-prime factor). Plates that passed the quality control (Z'>0.5), were then further analyzed using GeneData Screener software applying Levenberg-Marquardt algorithm (LMA) for curve-fitting strategy. Fitting strategy was considered acceptable if $R^2 > 0.8$.

## Animals and housing

These experiments and procedures were reviewed and approved by the United States Army Medical Research Institute for Infectious Diseases Institutional Animal Care and Use Committee. All research was conducted in compliance with the United States Department of Agriculture Animal Welfare Act (PHS Policy) and other federal statutes and regulations relating to animals and experiments involving animals and adhered to the principles stated in the *Guide for the Care and Use of Laboratory Animals*, National Research Council, 2011. The facility is fully accredited by the Association for Assessment and Accreditation of Laboratory Animal Care, International. All efforts were made to minimize pain and distress of study animals.

Seventy-eight female BALB/c mice, between 6 and 8 weeks old at time of virus exposure, were obtained from Charles River and randomized in six study groups. Sixteen adult male Indian-origin rhesus macaques (*Macaca mulata*) (RMs) between 5.1 and 5.6 years old at the time of virus exposure were provided by Merck & Co., Inc., Rahway, NJ, USA for this study. The RMs were randomized into 4 study groups. Ages and average body weights for RMs in each study group were similar. Although RMs used on this study were not research naïve, they had not been used on any prior study involving a filovirus, filovirus vaccine candidate, or filovirus therapeutic product. To ensure only appropriate animal research subjects were included in the study, all animal records were thoroughly reviewed by a veterinarian prior to shipment, and all animals were deemed healthy, free of pathogens that could interfere with study results, and appropriate for use in this study.

Mice and RMs were acclimated in animal biosafety level (ABSL)-4 animal rooms for at least 7 days prior to virus exposure (Day 0). Animal rooms were lit with fluorescent lights on a 12-hour light/dark cycle, except when interrupted for study activities (i.e., health status checks when animals were critically ill). RMs were housed individually, and mice were housed in groups of 10. Animals were provided with species appropriate Teklad Global diets (Harlan Teklad), fruits (as appropriate), and water *ad libitum* via an automatic watering system or water bottles. Enrichment was provided regularly as recommended by the *Guide for the Care and Use of Laboratory Animals*. In addition, RMs were provided with two servings of BioServ ElectroGel daily starting on the day that at least one animal displayed signs of disease. ElectroGel was discontinued when all animals were deemed healthy. For mice, most scheduled study procedures were performed without the use of anesthesia. Scheduled study procedures involving manipulation of RMs, including phlebotomy, physical examination, MSD-914/vehicle administration, and virus administration were performed while the animals were under anesthesia (8–12 mg/kg Ketamine HCl).

## Animal observations—RMs

Starting on Day 0, RMs were observed cage side at least once daily while healthy, and up to five times daily (at appropriately spaced intervals) when clinical signs of illness based on responsiveness scores defined in **Table 1** were evident. Observations included an assessment for cough, edema, rash, bleeding, and motor function.

Other observations (such as biscuit/fruit consumption, condition of stool, and urine output) were documented when possible.

**Table 1. Responsiveness scores.**

| Score | Criteria |
|-------|----------|
| 0 | Alert, responsive, normal species-specific behavior |
| 1 | Slightly diminished general activity, subdued but responds normally to external stimuli |
| 2 | Withdrawn, may have head down, upright fetal posture, hunched, reduced response to external stimuli |
| 3 | Prostrate but able to rise if stimulated, or dramatically reduced response to external stimuli |
| 4 | Persistently prostrate (unable to rise when stimulated), or severely or completely unresponsive, or continuous seizure |

Physical examinations were performed any time an animal was anesthetized for a study procedure (i.e., phlebotomy, treatment, virus exposure, and/or euthanasia). During the exams, animals were assessed for signs of disease and rectal temperatures and body weights were measured.

Animals that met the following pre-defined criteria were deemed moribund and euthanized: Responsiveness Score = 4 or Responsiveness Score = 3 and rectal temperature ≤34°C, or Responsiveness Score = 3 and two or more chemistry values outside of range (Blood urea nitrogen [BUN] ≥68 mg/dL, calcium [CA] ≤6.8 mg/dL, gamma-glutamyl transferase [GGT] ≥391 U/L, and/or creatinine [CRE] ≥2.8 mg/dL).

Moribund and end-of-study (Day 10) RMs were euthanized by intracardiac administration of a pentobarbital-based euthanasia solution under deep anesthesia in accordance with current *American Veterinary Medical Association Guidelines on Euthanasia* and USAMRIID standard operating procedures.

## Animal observations—mice

Mice were observed at least once daily starting on Day 0 postexposure (PE) while healthy, and two times daily (at appropriately spaced intervals) when clinical signs of illness (dull/rough coat, hunched posture, less mobile, decreased alertness) were evident. Group mean body weights were also obtained once daily. Mice were deemed moribund and humanely euthanized if they were unresponsive when provoked. Three mice from each group were humanely euthanized on Day 6, and blood collected and processed for RT-qPCR.

Moribund, scheduled Day 6, and end-of-study (Day 21) mice were euthanized by exsanguination followed by cervical dislocation under deep anesthesia in accordance with current *American Veterinary Medical Association Guidelines on Euthanasia* and USAMRIID standard operating procedures.

## Virus and virus exposure

The stocks of EBOV and mouse-adapted EBOV (maEBOV) have been described previously [4,11]. On the day of virus exposure, Day 0, animals were exposed to a target dose of 100 plaque forming units (pfu) of either EBOV administered by the IM route (RMs) or maEBOV administered by the IP route (mice). A sample of the exposure material was titrated by neutral red plaque assay as described previously [21] to determine the actual injected pfu.

## MSD-914 preparation and administration

MSD-914 in powdered form was reconstituted using 0.5% methylcellulose within 4 days prior to administration to achieve a solution with a final concentration of 0.5 mg/ml (for mouse studies) and 0.16 mg/mL (RM studies). Additional dilutions were made in 0.5% methylcellulose as necessary. Dilutions were stored at ambient temperature overnight or were held at 2–8°C for up to three days. When cold storage was employed, the dilutions were removed from cold storage and stirred until ambient temperature was achieved prior to dosing. All MSD-914, or vehicle (0.5% methylcellulose for mice and RM controls) treatments were administered via Oral gavage SID starting after virus exposure on Day 0 and continuing through

Day 8. For RM, administration of MSD-914 or vehicle was followed by a hydrated monkey chow slurry to minimize risk of regurgitation and help promote absorption.

## Blood collection, clinical pathology, RT-qPCR, and serum plaque assay

Blood was collected via venipuncture. Processing of blood samples for hematology and clinical chemistry analysis have been described previously [11]. For hematology, a Beckman Coulter DxH520 hematology analyzer was used. For clinical chemistries, Abaxis Piccolo blood chemistry analyzers with General Chemistry 13 panels were used. Extraction of RNA and RT-qPCR was performed as described previously [11]. The neutral red plaque assay described in Shurtleff, 2016, was used for the titration of virus in serum with one modification [10]. Multiple serum samples obtained on Days 5 and 7 and at terminal time points were slightly gelatinous following thawing, so an initial 1:2 dilution in PBS was performed prior to further dilutions and/or plating.

## Bioanalytical analysis of plasma MSD-914 levels

All samples used in pharmacokinetic analysis were inactivated using a protein-precipitation method approved for viral inactivation prior to removal from biocontainment. K2EDTA plasma samples collected for PK analysis were treated with formic acid (6% v/v final volume ratio), and 50 µL of this mixture was treated with 25 µL of an internal standard working solution (ISWS) containing 200 ng/mL of internal standard (DAPDA) in 50:50 water:methanol. Samples were then treated with 900 µL of 50:35:15 acetone:methanol:water, inverted several times, and allowed to stand for at least 15 min. Samples were centrifuged (14,000 g, 15 min), and the supernatant was dried under a stream of nitrogen. Samples were reconstituted in 175 µL of 95:5 acetonitrile:water, mixed, and centrifuged (2,000 g, 5 min). Calibration standards and quality control samples were treated identical to study samples. All samples were analyzed by LC/MS-MS using an AB Sciex 5500 Triple Quad mass spectrometer in line with a Shimadzu Prominence UPLC system. Sample separation was performed with a Synergi Hydro-RP column (Phenomenex, 30 x 2 mm, 4 µm, 80 Å) using a gradient of mobile phase B (acetonitrile + 0.1% formic acid) into mobile phase A (water + 0.1% formic acid) over 3 min. Analytes were detected by positive electrospray ionization through multiple reaction monitoring of both MSD-914 (253.1 → 153.1 DP = 61 V, CE = 33 V, CXP = 8 V) and DAPDA (267.1 → 151.1, DP = 61 V, CE = 23 V, CXP = 10 V). The upper limit of quantitation (ULOQ) of MSD-914 in NHP plasma was defined as 500 ng/mL, while the lower limit of quantitation (LLOQ) was defined as 0.5 ng/mL.

## Determination of plasma free fraction (plasma protein binding)

The plasma free fraction of the test compound was outsourced to $Q^2$ solutions (Durham, NC) and determined by equilibrium dialysis. Equilibrium dialysis was performed using a 96-well equilibrium dialysis device and a dialysis membrane with a 12–14 kDa molecular weight cut-off (HTDialysis, Gales Ferry, CT). Briefly, the dialysis membranes were prepared, and the equilibrium dialysis device were assembled according to manufacture instruction (http://www.htdialysis.com/). Aliquots of plasma containing test compound at 2.5 µM concentration were added to one side of the dialysis membrane and phosphate buffer saline (100 mM, pH 7.4) was added to the other side of the dialysis membrane. The equilibrium dialysis plate with the samples was incubated for 4 hour at 37°C under 5% $CO_2$. Following incubation, 50 µL of plasma and buffer were removed from each dialysis well, a matrix match was added to bring the pre-extracted samples to 50% plasma and 50% buffer, and the analyte was extracted by the addition of 250 µL acetonitrile containing 200 nM of the internal standard mixture (labetalol, imipramine, and diclofenac). The samples were vortex-mixed and centrifuged for 5 min at 3200 g, and the resulting supernatant fractions were analyzed by liquid chromatography-tandem mass spectrometry (LC-MS/MS). Six replicates (n = 6) were used for each compound. The free (unbound) fraction of test compounds in the plasma was determined by dividing the compound peak area over the internal standard area in the buffer by that in the dialyzed plasma samples.

## Pharmacokinetics and tolerance evaluation

Uninfected male mice were used to assess the pharmacokinetics (PK) of MSD-914. The mice were administered the test compound orally via oral gavage at doses of 0.31, 1.25, and 5 mg/kg (n = 3; male). The compound was formulated as described above. Blood samples were collected from the femoral vein or artery in tubes containing K2EDTA as an anti-coagulant. After centrifugation, plasma samples were obtained from each mouse over a 24-hour period. Plasma samples were collected at 0.25, 0.5, 1, 2, 4, 8, 16, and 24 hours post-dose.

To assess the tolerance of MSD-914 in RM, two uninfected male RMs were given the test compound orally via oral gavage for 10 consecutive days as a single dose. The compound was formulated as described above. The administration consisted of 0.2 mg/kg/day from Day 1 to Day 3, 0.8 mg/kg/day from Day 4 to Day 6, and 3 mg/kg/day from Day 7 to Day 10. Clinical observations were conducted immediately after dosing and once daily up to 13 days post-dose. Plasma samples were collected on Day 3, Day 6, and Day 10 as described above to evaluate the PK of the three dose regimens. Plasma samples were obtained at 0.25, 0.5, 1, 2, 4, 8, and 24 hours post-dose.

All samples from mice and RMs were subject to analysis, and the levels of MSD-914 were measured using the bioanalytical method described above. The plasma drug level data were then analyzed using Phoenix WinNonlin (version 8.0) to perform non-compartmental analysis.

## Necropsy, histology, and immunohistochemistry

Necropsies on RMs were performed by a board-certified pathologist. Tissue sections were trimmed, fixed in neutral buffered formalin, and held in biocontainment for 21 days prior to further processing. The sections were then removed from biocontainment, routinely processed, embedded in paraffin, and cut to 5 µm thick. For histology, slides were deparaffined, stained with hematoxylin and eosin, and cover-slipped. For immunohistochemistry, replicate sections were placed on positive-charged slides and stained using a mouse monoclonal anti-EBOV antibody.

## Statistical analysis

The survival rates at study termination days were compared by Fisher's exact test, and the times to death were analyzed by the log-rank test for the pairwise comparison between the groups. Hematology, clinical chemistry, and RT-qPCR group comparisons were performed by Friedman tests. Comparisons by day, through Day 7 PE, were performed by Wilcoxon rank sum tests. Analyses were performed using SAS version 9.4.

## Results

### Anti-filovirus activity of MSD-914

*In vitro* testing of MSD-914 exhibited potent antiviral activity against multiple strains of Filoviruses and in multiple cell types. The $EC_{50}$ value ranged from 30–65 nM when tested in HeLa, HFF and MRC-5 cells infected with EBOV. MSD-914 exhibited $EC_{50}$ of 71 nM and 55 nM in HeLa cells infected with MARV and SUDV respectively (**Table 2**). The selectivity index was > 100 in all the *in vitro* screens. Overall, the results from the *in vitro* screen suggest a potent broad-spectrum anti filovirus activity of MSD-914.

### Oral administration of MSD-914 protects mice following maEBOV exposure

Seventy-eight female BALB/c mice were randomized into six study groups, each containing thirteen mice (**Table 3**). Group 1 was the control group and received PBS. Groups 2–6 received MSD-914 at dose levels ranging from 5 mg/kg down to 0.02 mg/kg. Treatments were administered PO SID Day 0 (within one hour post-virus exposure) through Day 8 postexposure (PE). All mice were exposed IP to 100 pfu of maEBOV on Day 0. They were observed daily for signs of disease and were euthanized when moribund. Three animals from each group were euthanized on Day 6 PE, and blood was collected for RT-qPCR. Animals that survived until end-of-study were euthanized on Day 21 PE.

                                  

**Table 2. *In vitro* potency of MSD-914 against EBOV, MARV and SUDV.**

| Cell Type | Pathogen | EC50 [µM] | EC90 abs [µM] | CC50 [µM] | SI |
|---|---|---|---|---|---|
| MRC-5 | EBOV | 0.027 | 0.35 | >30 | >1111 |
| HeLa | EBOV | 0.064 | 0.43 | >25 | >390.6 |
| HFF | EBOV | 0.065 | 0.50 | >25 | >384.6 |
| HeLa | MARV | 0.071 | 0.58 | >25 | >352.1 |
| HeLa | SUDV | 0.055 | 1.11 | >25 | >454.5 |

**Table 3. Group designations—Mice.**

| Group | Test/Control article | Dose (mg/kg) | Treatment schedule (Days PE) | Route | N= |
|---|---|---|---|---|---|
| 1 | 0.5% methyl cellulose | N/A | 0–8 | Oral | 13* |
| 2 | MSD-914 | 5 | | | 13* |
| 3 | | 1.25 | | | 13* |
| 4 | | 0.31 | | | 13* |
| 5 | | 0.08 | | | 13* |
| 6 | | 0.02 | | | 13* |

*On Day 6 postexposure (PE), three animals per group were euthanized, and blood was collected for RT-qPCR. All other animals were euthanized when moribund or on Day 21 PE.

Percent survival is shown in **Fig 1** for the 10 animals in each group that did not undergo scheduled euthanasia on Day 6. All animals in Groups 2–4 (5, 1.25, and 0.31 mg/kg, respectively) survived until end-of-study (Day 21 PE). Four out of 10 (40%) of the animals in Group 5 (0.08 mg/kg) survived until end-of-study. Survival for animals in Groups 1 (controls) and 6 (0.02 mg/kg) was the same, with 1/10 (10%) of the animals surviving until end-of-study.

All control animals, and all animals in Groups 5 (0.08 mg/kg) and 6 (0.02 mg/kg) were ill on one or more days PE, and weight loss of 12% to 20% was noted for these groups (**Fig 2**). Three animals (30%) in Group 2 (5 mg/kg) also became ill during the course of the study (**Fig 1**), but none of the animals in Groups 3 (1.25 mg/kg) and 4 (0.31 mg/kg) became ill at any time point. Weight loss was not noted for any of these 3 groups (**Fig 2**).

The total amount of viral RNA present in each group of mice six days post exposure was measured. RT-qPCR was performed on EDTA plasma samples for the three animals per group euthanized on Day 6 PE (**Fig 3**). Surprisingly, viral RNA at comparable levels was detected for all groups. The presence of viral RNA indicates that all groups were viremic on Day 6 PE. RT-qPCR does not distinguish between live/infectious virus or dead/noninfectious virus (or genomic material). Therefore, the data only indicate that EBOV or EBOV genomic material at comparable levels was present in the blood of these animals on Day 6 PE. Plaque assay to assess infectious virus levels was not performed in this study.

## Pharmacokinetics evaluation in mice

The pharmacokinetics of MSD-914 was investigated in mice to explore the relationship between exposure and efficacy. Due to the requirement of an animal biosafety level 4 (ABSL4) laboratory for EBOV-infected animal studies, uninfected animals were selected for the pharmacokinetic (PK) study to avoid methodological and technical limitations.

Plasma protein binding assays were conducted prior to PK evaluation and revealed a high degree of unbound fraction for MSD-914 (80% in mouse and 93.6% in RM). This suggests that the measured drug concentration in PK plasma samples is approximately equivalent to the unbound drug concentration in plasma.

PK evaluation in mice was carried out at doses of 0.31 mg/kg, 1.25 mg/kg, and 5 mg/kg (**Fig 4**) as these three dose regimens demonstrated improved survival in a mouse-adapted EBOV model. All doses were administered orally using the same vehicle (0.5% methylcellulose). All samples were analyzed, and drug concentrations were measured using liquid

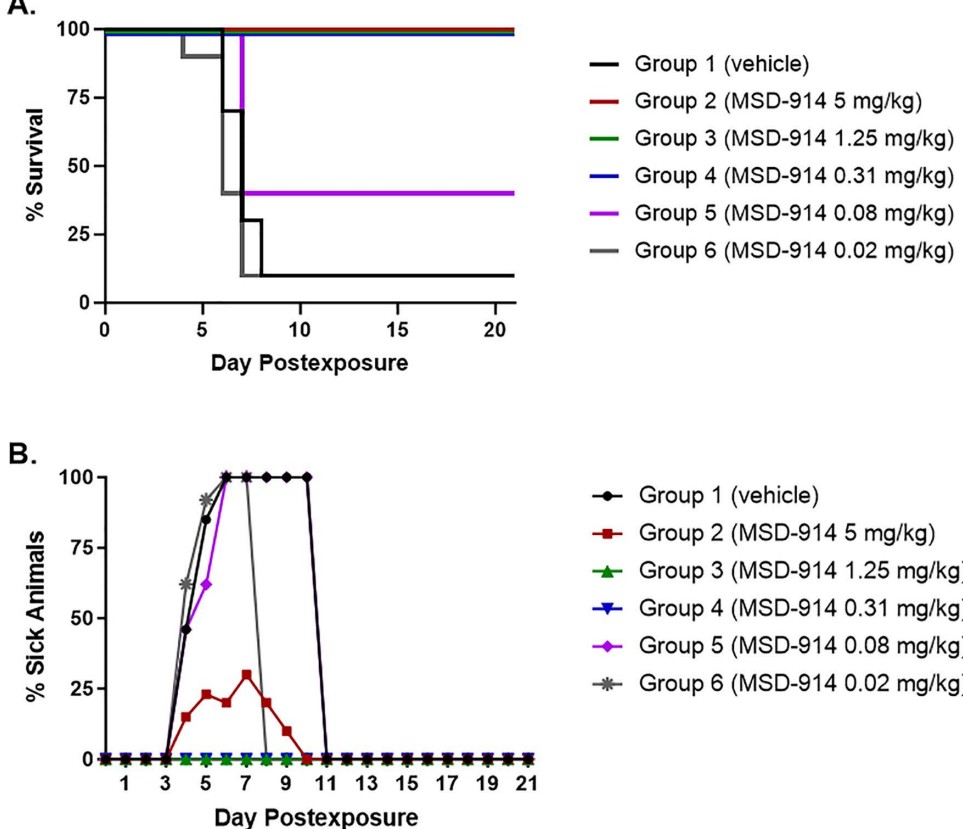

**Fig 1. Morbidity and mortality in maEBOV-exposed mice treated with MSD-914. (A)** Percent survival is shown for the 10 animals in each group that did not undergo scheduled euthanasia on Day 6. **(B)** The percent of animals in each group that were sick was determined by dividing the number of animals that were sick by the total number of animals remaining in a group on a particular study day, and then multiplying by 100 to determine the percentage of sick animals.

chromatography-tandem mass spectrometry (LC-MS/MS). Pharmacokinetic parameters were calculated using noncompartmental modeling (**Table 4**).

It was observed that the exposure of MSD-914, as determined by the maximum concentration of drug in plasma ($C_{max}$) and the area under the concentration-time curve from time zero to the last measurable concentration ($AUC_{0-t}$), increased with dose level. Non-linearity in MSD-914 exposure across all three dose regimens was observed. The plasma concentration in the 0.31 mg/kg dose regimen constantly remained below the *in vitro* $EC_{50}$ value (65 nM) while the plasma concentrations in the other two dose regimens achieved coverage of the in vitro $EC_{50}$ for 1 hr (1.25 mg/kg) and 8 hr (5 mg/kg). This suggests that sustained plasma concentrations above the *in vitro* $EC_{50}$ were not associated with improved survival and were not necessary for *in vivo* efficacy.

**RM drug tolerability study to determine maximum dose**

The ability of MSD-914 to completely protect mice from mortality following maEBOV infection at doses as low as 0.31 mg/kg was encouraging and supported proof-of-concept (POC) evaluation in nonhuman primates.

Prior to initiating the RM POC study, a 10-day oral administration ascending dose (0.2 mg/kg/day for Days 1–3, 0.8 mg/kg/day for Days 4–6, and 3 mg/kg/day for Days 7–10) tolerability and toxicokinetic study in two rhesus macaques (RMs)

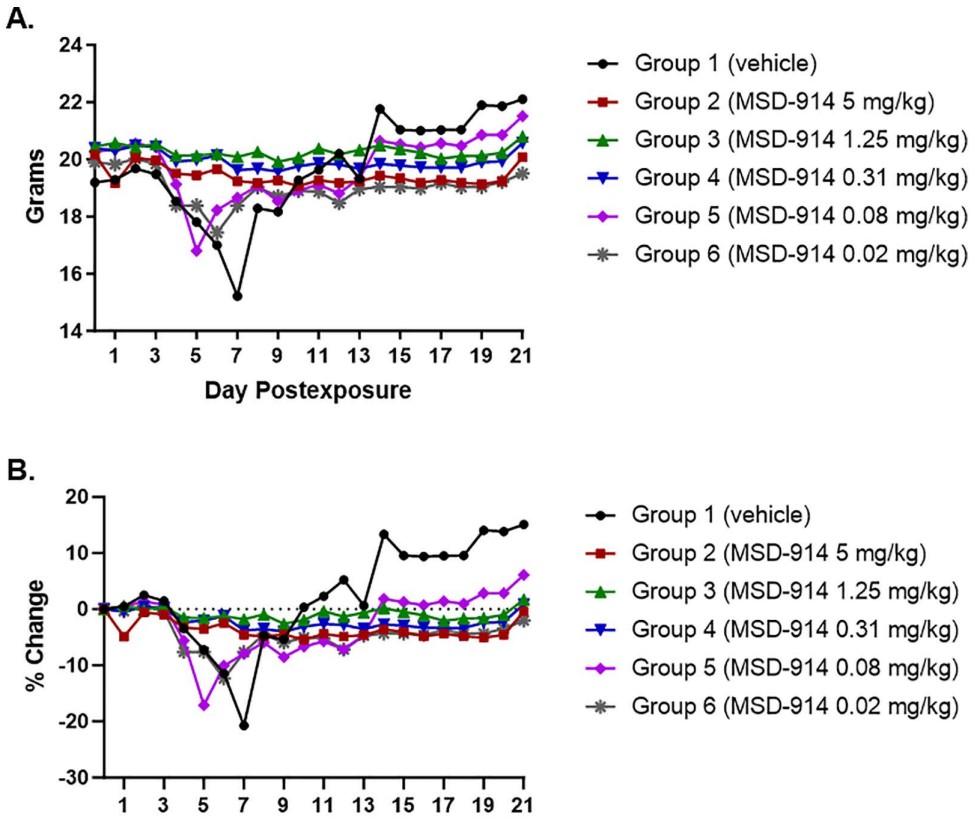

**Fig 2. Effect of MSD-914 treatment on weight loss in mice following ma-EBOV exposure. (A)** The average body weight for each group was determined by dividing the group body weight by the total number of animals remaining in a group on a particular study day. **(B)**. Percent change in average group body weights was determined by subtracting the baseline average group body weight from the average group body weight for a particular study day, dividing the resulting value by the baseline average group body weight, and then multiplying by 100 to generate the percentage.

was conducted to inform on appropriate dose levels in RM (**Table 5**). Tolerability was based on unscheduled deaths, clinical observations, body weights, food consumption, and clinical pathology evaluations (limited to serum biochemistry). There were no unscheduled deaths or drug-related changes in body weight or food consumption during the study, and there were no drug-related clinical signs observed through Day 6. Administration of 0.8 mg/kg/day was associated with a very slightly increased alanine transaminase (ALT) and aspartate aminotransferase (AST) in one animal on Day 6. Administration of 3 mg/kg/day of MSD-914 exceeded the maximum tolerated dose for RMs, and dosing was suspended following the third daily dose of 3 mg/kg due to adverse clinical signs. Drug-related effects included observations of decreased activity, emesis, and pale appearance on Day 10. This same animal had markedly decreased glucose, markedly increased ALT, and moderately increased AST, bilirubin, and direct bilirubin. Slightly increased gamma-glutamyl transferase (GGT) and triglycerides, and slightly decreased sodium, potassium, and chloride were also noted. The other animal had slightly increased ALT and triglycerides and slightly decreased potassium on Day 10.

Similar to the PK in mice, exposure of MSD-914 in RMs, as determined by $C_{max}$ and $AUC_{0-t}$, increased with the dose level in a non-linear pattern. The RM PK profile shows that plasma concentrations following the 0.2 mg/kg and 0.8 mg/kg dose regimens achieved coverage of the *in vitro* $EC_{50}$ for 2 hr (0.2 mg/kg) and around 4 hr (0.8 mg/kg). These exposures represent superior coverage over the *in vitro* $EC_{50}$ when compared with those obtained from the efficacious doses in mice

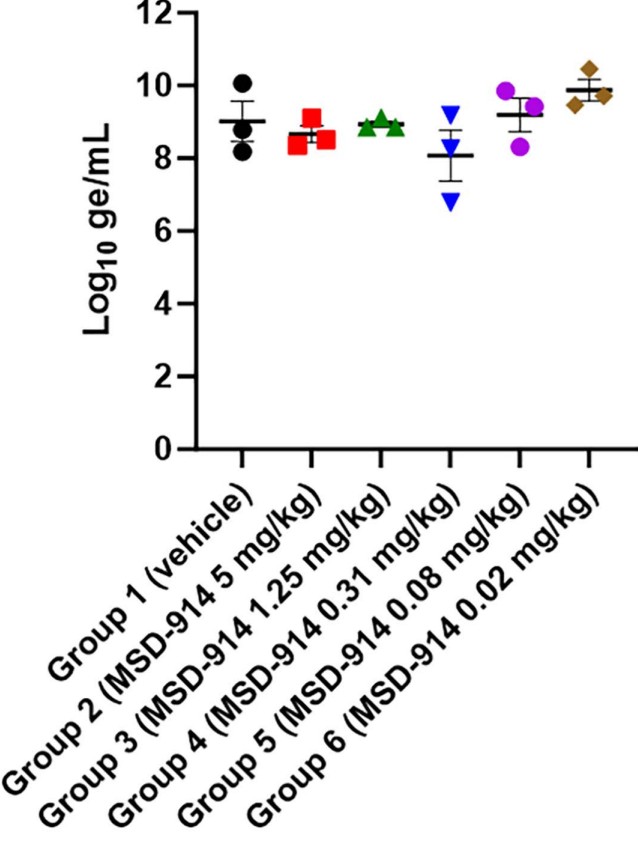

**Fig 3. Viral RNA in plasma from maEBOV-exposed mice treated with MSD-914.** RT-qPCR was performed for three animals per group that were euthanized on Day 6 postexposure. Symbols represent individual animals, with the mean per group depicted by the horizontal line. Error bars represent the standard error of the mean (SEM).

(0.31 mg/kg and 1.25 mg/kg) (**Fig 4**). The $AUC_{0-t}$ from the 0.8 mg/kg dosing group in RMs was also 2.4-fold higher than the $AUC_{0-t}$ of the efficacious dose in mice (1.25 mg/kg) (**Table 5**).

Based on the results of the tolerability and toxicokinetic study, and the PK evaluation of mice and RM, the maximum tolerated dose of MSD-914 delivered orally in RMs was determined to be 0.8 mg/kg/day and this dose should provide adequate exposure to achieve in vivo efficacy in RM.

### No protection was afforded by MSD-914 in EBOV-infected Rhesus macaques

The POC study included sixteen adult RMs randomized into 4 study groups, balanced by body weight and each containing 4 RMs (**Table 6**). Groups 1, 2, and 3 received 0.8 mg/kg/day (Group 1), 0.27 mg/kg/day (Group 2), or 0.09 mg/kg/day (Group 3) of MSD-914. Group 4 served as the vehicle control group and was administered 0.5% methylcellulose. RMs were acclimated to their ABSL-4 housing for over a week prior to challenge. On Day 0, RMs were challenged IM with 135 pfu of EBOV. Treatment with MSD-914 occurred via oral gavage SID on Days 0–10 PE (on Day 0, treatment was initiated within 1 hour of virus exposure).

All animals on study developed clinical signs of EBOV disease (**Table 7**, **Fig 5**, and **S1 Fig**) including lymphadenopathy, rash, fever (≥1.5°C above baseline, with baseline defined as the average of data from Day −4 and Day 0 for an animal), facial swelling, motor dysfunction, and discharge/bleeding. Terminal hypothermia (≥2°C below baseline) was

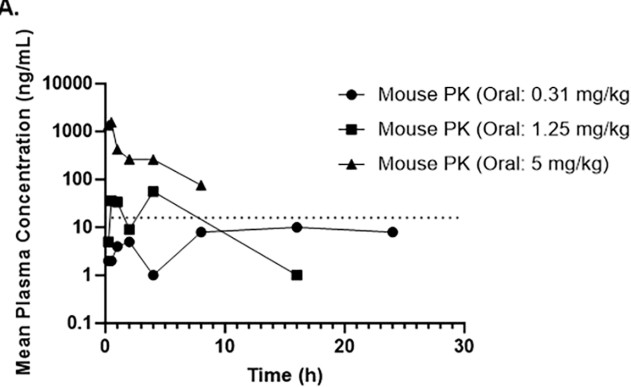

**A.**

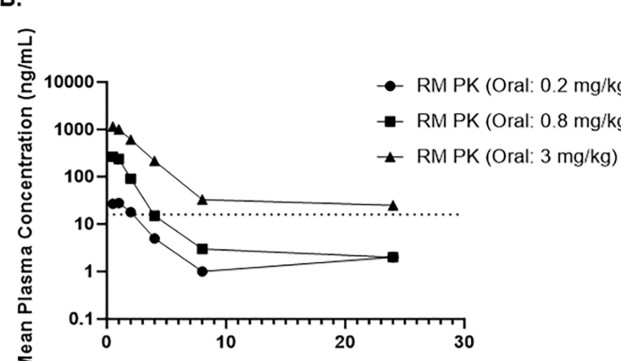

**B.**

**Fig 4. MSD-914 pharmacokinetics.** (A) Pharmacokinetics following oral administration in uninfected mouse (mean, n = 3). The dotted horizontal line represents the EC50 in HeLa cells. (B) Pharmacokinetics following oral administration in uninfected RM (mean, n = 2). The dotted horizontal line represents the EC50 in HeLa cells.

**Table 4. Mean pharmacokinetics data in mice and RMs treated with MSD-914.**

| Dose (mg/kg) | Species | Administration | Cmax (ng/ml) | $AUC_{0-t}$ (hr*ng/ml) |
|---|---|---|---|---|
| 0.31 | Mice | Oral | 10.7 | 175.8 |
| 1.25 | Mice | Oral | 55.5 | 223.8 |
| 5 | Mice | Oral | 1562.8 | 2576.4 |
| 0.2 | RM | Oral | 28.2 | 98.8 |
| 0.8 | RM | Oral | 264.6 | 531.7 |
| 3 | RM | Oral | 1149.1 | 3150.0 |

**Table 5. Maximum tolerated dose in RM.**

| Day | Dose (mg/kg) | No. of animals | Survival (no. of RM) | Clinical observations(s) |
|---|---|---|---|---|
| 1-3 | 0.2 | 2M | 2 of 2 | none |
| 4-6 | 0.8 | 2M | 2 of 2 | none |
| 7-10 | 3 | 2M | 2 of 2 | nasal discharge, decreased activity |

**Table 6. Group designations-Rhesus macaques.**

| Group | Test/Control article | Dose (mg/kg) | Treatment schedule (Days PE) | Route | N= |
|---|---|---|---|---|---|
| 1 | MSD-914 | 0.8 | 0–10 | Oral | 4 |
| 2 | | 0.27 | | | 4 |
| 3 | | 0.09 | | | 4 |
| 4 | Vehicle (0.5% methylcellulose) | N/A | | | 4 |

PE = postexposure.

**Table 7. Clinical observations for Rhesus Macaques (RM).**

| Group | 1 | | | | 2 | | | | 3 | | | | 4 | | | |
|---|---|---|---|---|---|---|---|---|---|---|---|---|---|---|---|---|
| RM # | 2 | 4 | 10 | 15 | 1 | 6 | 13 | 14 | 3 | 5 | 12 | 16 | 7 | 8 | 9 | 11 |
| Fever[1] | X | | X | X | X | | | | X | | X | | X | | X | |
| Hypothermia[2] | X | | | X | | X | | X | | X | | X | X | | | |
| EBOV Rash | X | X | X | X | X | X | X | X | X | X | X | X | X | X | X | X |
| Lymphadenopathy | X | X | X | X | X | X | X | X | X | X | X | X | X | X | X | X |
| Motor Dysfunction | | X | X | | X | | | X | X | X | X | X | X | X | X | X |
| Facial Swelling | | X | X | | | | | | X | X | X | X | X | X | | X |
| Nasal Discharge | | | X | | | | | | X | X | X | | | | X | X |
| Ocular Discharge | | | X | | X | | | | | | | | X | X | X | |
| Bleeding – Nose/Mouth/ Vomit | | | | | | | | X | X | X | | X | | | | |
| Bleeding – Penis | | | | | | X | | | X | | | | | | X | |
| Bleeding – Anus/Stool | | | | | X | | | | X | | X | | X | | | |
| Anorexia[3] | | | | | X | | | | X | | X | X | X | | X | X |
| Palpable GI Fluid | | | | X | X | | | X | X | X | X | | | | | |
| Weight Loss[4] | | | | | | X | X | | X | | X | | | | X | |
| Cyanosis | | | X | | X | | | | X | | | | | | | |
| Loose/soft Stool | X | | | | | | | | X | | | | X | | | |
| No Stool | | | | | | | | | | | | | X | | | X |

[1]Defined as a temperature greater than or equal to 1.5°C above baseline; [2]Defined as a temperature less than or equal to 2°C below baseline; [3]Defined as an absence of biscuit and enrichment consumption for one or more days OR an absence of biscuit consumption for 3 or more consecutive days; [4]Loss of greater than 0.2 kg compared to baseline.

noted for about half of the animals on study, and anorexia was commonly observed for Group 3 (0.09 mg/kg) and control animals. Weight loss, as defined as a loss of greater than 0.2 kg compared to baseline was noted for half of the animals in Groups 2 and 3, and one animal in Group 4 (control animals). To assess differences in the magnitude of rectal temperature changes and weight loss (**Fig 5**), a baseline was determined for each parameter by averaging the values from Day −4 and Day 0 by animal, and then adjusting peak values (i.e., the highest values and lowest values for rectal temperature, and the lowest values for body weight for each animal) by determining the "peak percent (%) change from baseline" using the 100 × (Peak Value − Baseline)/Baseline.

Peak percentage change from baseline for body weights and rectal temperatures were not significantly different between groups.

In total, 13/16 animals reached a moribund state prior to study termination (**Fig 5**). On Day 10 PE, the remaining two animals in Group 2 (0.27 mg/kg) were ill, with evidence of severe and progressing EBOV disease; it was anticipated that neither of these animals was going to survive. Although the remaining animal in Group 1 (0.8 mg/kg) had a

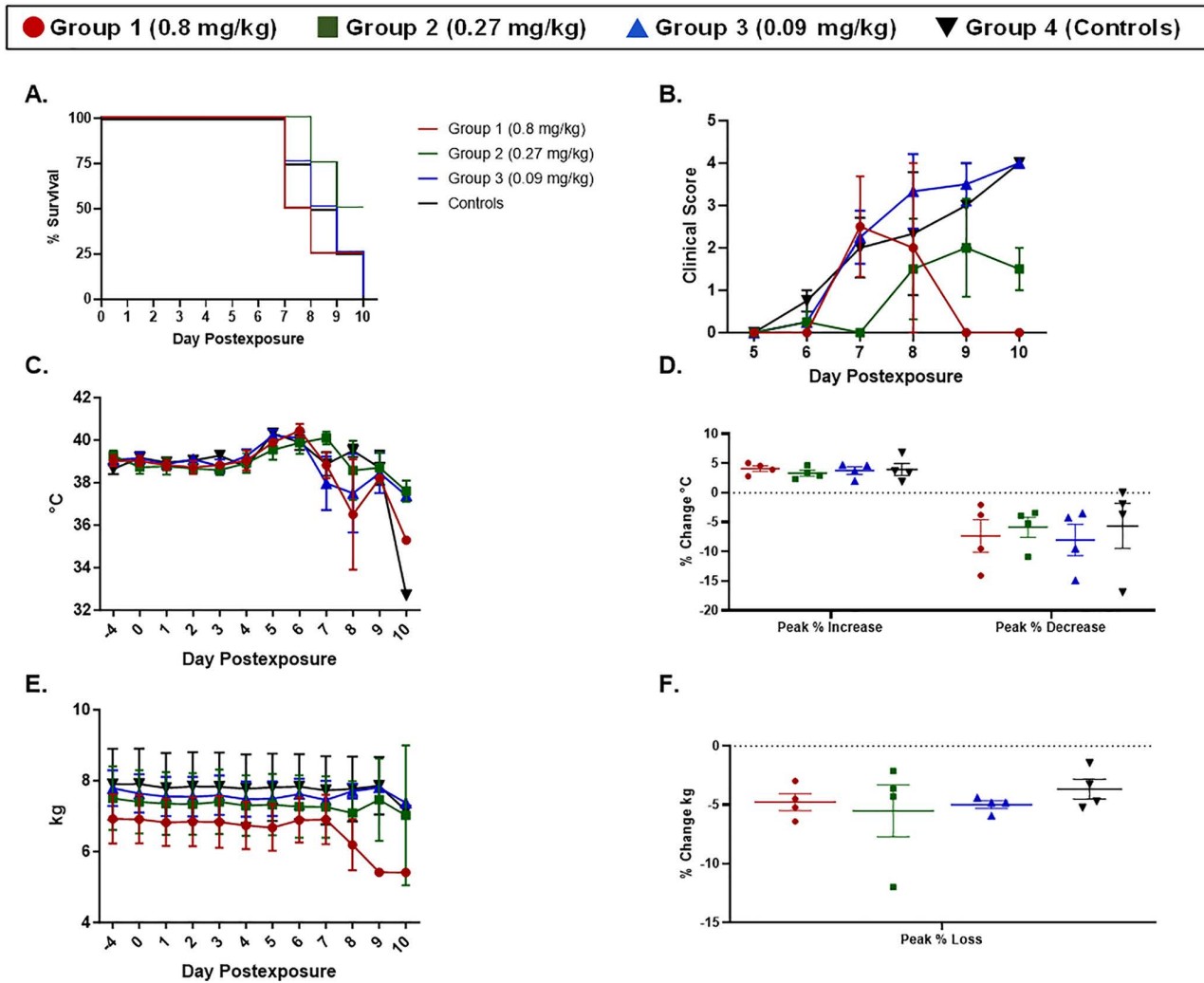

**Fig 5. Morbidity and mortality in EBOV-exposed RMs treated with MSD-914.** Red lines and symbols represent Group 1 (0.8 mg/kg), green lines and symbols represent Group 2 (0.27 mg/kg), blue lines and symbols represent Group 3 (0.09 mg/kg), and black lines and symbols represent Group 4 (controls). Error bars represent the SEM. **(A)** Percent survival. Note that the remaining animals in Groups 1 and 2 were euthanized on Day 10 PE due to welfare concerns, but that they were not yet moribund. Therefore, the survival line for these two groups does not include the euthanasia of those three animals. **(B)** Clinical score. Shown is the group mean per Day PE. The clinical score is the responsiveness score with the following clarifications: 1) An animal with a responsiveness score of 3 that met criteria for euthanasia based on clinical chemistry parameters was given a clinical score of 4; 2) An animal found deceased was given a clinical score of 5. **(C)** Change in temperature over time – group mean per Day PE. **(D)** Percent change from baseline for the highest and lowest temperature values. Symbols represent individual animals, and the horizontal line is the group mean. **(E)** Change in weight over time – group mean per Day PE. **(F)** Percent change from baseline for the lowest measured body weight. Symbols represent individual animals, and the horizontal line is the group mean.

responsiveness score of 0 on Day 10 PE, clinical chemistry data revealed that this animal did have severe and potentially progressing EBOV disease. This data implies a lack of efficacy in RMs and the decision was made to terminate the study on Day 10 PE.

Mean time to death (MTD) was calculated by subtracting time of infection from time of euthanasia or observed found deceased. MTD calculations did not include the three animals euthanized on Day 10 PE as these animals did not meet the criteria to be considered moribund prior to euthanasia. The MTD for control animals was 8.64 days. The MTD for

Groups 1 (0.8 mg/kg), 2 (0.27 mg/kg), and 3 (0.09 mg/kg) was 7.85, 8.74, and 8.45 days, respectively, and MTDs were not significantly different from each other nor the control group.

## Clinical pathology and RT-qPCR confirm active EBOV infection in MSD-914-treated RMs

Hematology analysis was performed using a Beckman Coulter DxH520 hematology analyzer. Grouped data are shown in **S2 Fig**, with corresponding individual animal data shown in **S3 Fig**. Differences in the magnitude of a response (**S4 Fig**) were assessed using peak % change from baseline as described above.

Changes characteristic of EBOV infection were noted for animals on this study. Increases in total white blood cells (WBC) reflects primarily increased neutrophils (NE) and mobilization of surviving white cells in response to EBOV infection. Overall, Group 1 (0.8 mg/kg) appeared to have a more moderated response compared to other groups, and the % change from baseline for peak values for WBC and NE was significantly lower than the corresponding change for Group 2 (0.27 mg/kg). This is likely due to massive destruction of lymphocytes (LY) as depicted in **S2 Fig**. Group 2 was also unique in that WBC and NE remained elevated for a longer period of time compared to other groups. Whereas the other three groups had elevated values on Day 5 PE that returned to near baseline by Day 7 PE, Group 2 values remained elevated through Day 7 PE, returning to near baseline by Day 9 PE. LY and platelets (PLT) were notably reduced compared to baseline for all animals, and significant differences between groups for baseline-corrected values did not exist. A mild reduction in hematocrit (HCT) was also seen, suggestive of a RBC loss and/or RBC destruction. Significant differences between groups for baseline-corrected values did not exist.

Clinical chemistry analysis was performed using Abaxis Piccolo® blood chemistry analyzers. Grouped data are shown in **Fig 6**, with corresponding individual animal data shown in **S5 Fig**. Differences in the magnitude of a response (**Fig 7**) were assessed using peak percentage change from baseline as described above.

Similar to hematology, changes in clinical chemistry parameters characteristic of EBOV infection were noted for animals on this study. ALT, AST, and alkaline phosphatase (ALP) were significantly elevated above baseline, with peak values often obtained around the time animals became moribund. A primary target during EBOV infection in RMs is the liver, and the changes for these parameters are often indicative of active infection and subsequent tissue damage in the liver. Azotemia is also a hallmark of EBOV disease in RMs, indicated by elevated values for blood urea nitrogen (BUN) and creatinine (CRE) around the time animals become moribund. Characteristically, both BUN and CRE were elevated for animals on this study. Notably, the two animals in Group 2 (0.27 mg/kg) that were still alive on Day 10 PE had BUN and/or CRE values on Day 10 PE indicative of terminal disease, even though they had not yet met the criteria for euthanasia based on responsiveness score. The third animal still alive on Day 10 PE, from Group 1 (0.8 mg/kg), was not yet in a state of terminal disease based on clinical chemistry data but did have evidence of rapid disease progression based on BUN and CRE values. Based on the data, none of these animals would have likely survived another 12–24 hours, further justifying the decision for early study termination on Day 10 PE. For the most part, significant differences in baseline-corrected clinical chemistry parameters between groups were not noted with one exception. The peak percentage change from baseline for ALP was significantly greater for Groups 3 (0.09 mg/kg) and 4 (controls) compared to Group 1 (0.8 mg/kg) (**Fig 7**).

RT-qPCR (**Fig 8**) and serum plaque assays were performed to assess differences in viremia between groups. Viral RNA was detected in plasma as early as Day 3 PE, with peak titers noted on Day 7 PE and/or at the time animals became moribund. Although titers for Groups 1 (0.8 mg/kg) and 2 (0.27 mg/kg) were notably lower than titers for Groups 3 (0.09 mg/kg) and 4 (controls) on Day 5 PE, none of the titers were significantly different compared to controls. Average peak titers were 9.2, 8.1, 9.3, and 9.3 $Log_{10}$ genomic equivalents (ge)/mL for Groups 1, 2, 3, and 4, respectively. Again, Group 2 was notably lower than the other three groups, but this difference was not statistically significant. Animals alive on Day 10 PE (one from Group 1 and two from Group 2) had titers between 7.77 and 8.46 $Log_{10}$ ge/mL, indicating a significant amount of virus was present in circulation for these animals.

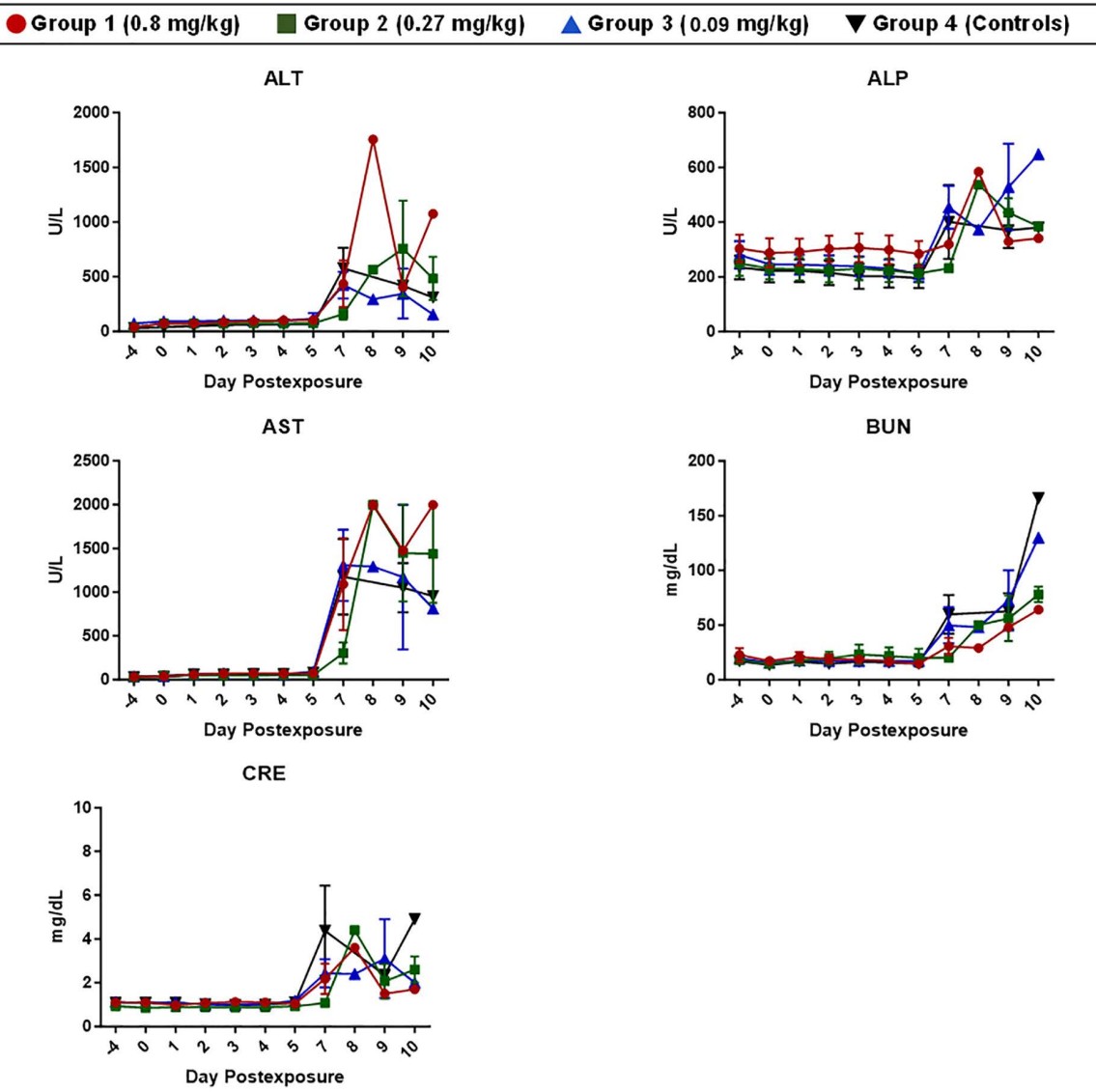

**Fig 6. Clinical chemistry changes in EBOV-exposed RMs treated with MSD-914.** Clinical chemistry was performed on Abaxis Piccolo° blood chemistry analyzers. Grouped data are shown, with error bars depicting the SEM. Red lines and symbols represent Group 1 (0.8 mg/kg), green lines and symbols represent Group 2 (0.27 mg/kg), blue lines and symbols represent Group 3 (0.09 mg/kg), and black lines and symbols represent Group 4 (Controls).

During peak EBOV disease, coagulation pathway activation can affect serum and/or plasma samples, resulting in fibrin clot formation. Fibrin clots were noted during processing of serum samples to be used for plaque assay on this study. Unfortunately, the presence of fibrin in the serum can affect the performance of the plaque assay, and as a result we were only able to identify infectious virus (at $2 \times 10^2$ pfu/mL) for one animal in Group 3 (0.09 mg/kg).

## Pathologic findings confirmed systemic EBOV disease for MSD-914-treated animals

Necropsies were performed on all animals in this study. Overall, findings were consistent with systemic EBOV disease. At necropsy, livers were pale and/or friable, and inguinal lymph nodes were enlarged and red. Red coloration was noted for

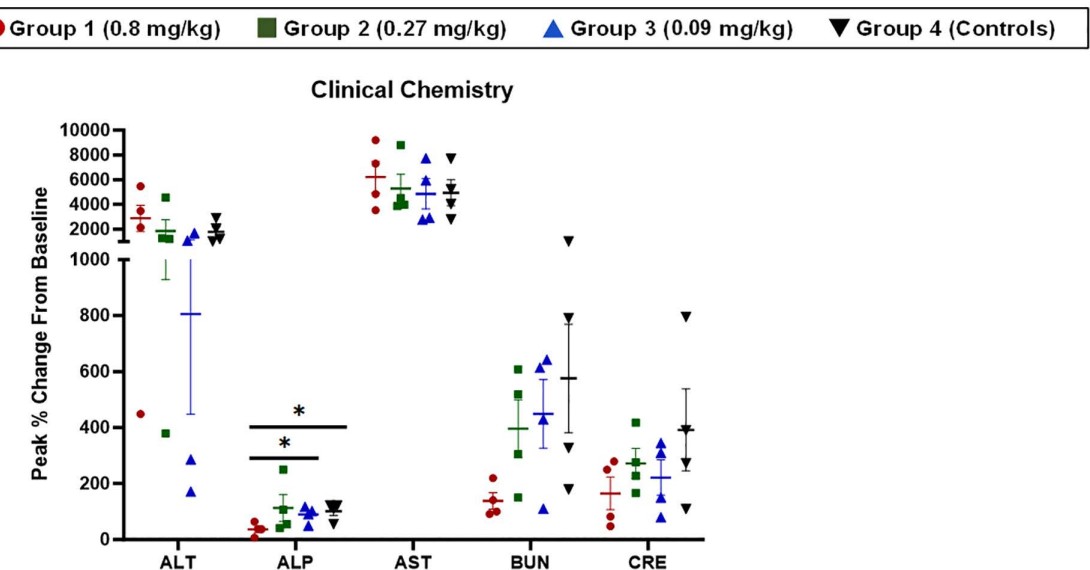

**Fig 7. Peak % change from baseline analyses for clinical chemistries.** The % increase from baseline for peak values is shown. Symbols represent individual animals, and the horizontal line is the group mean. Error bars represent the SEM. Statistical significance (p<0.05) is depicted by the *. Red lines and symbols represent Group 1 (0.8 mg/kg), green lines and symbols represent Group 2 (0.27 mg/kg), blue lines and symbols represent Group 3 (0.09 mg/kg), and black lines and symbols represent Group 4 (Controls).

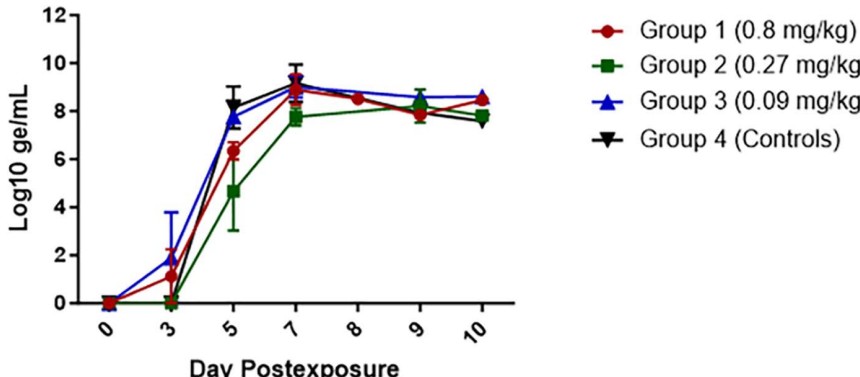

**Fig 8. Detection of EBOV RNA in plasma from MSD-914-treated RMs.** RT-qPCR was performed on an Applied Biosystems® 7500 Fast Dx instrument. Grouped data are shown, with error bars depicting the SEM. Red lines and symbols represent Group 1 (0.8 mg/kg), green lines and symbols represent Group 2 (0.27 mg/kg), blue lines and symbols represent Group 3 (0.09 mg/kg), and black lines and symbols represent Group 4 (Controls).

various other tissues, including adrenal glands, urinary bladder, small intestines, and stomach. The spleen was firm and/or friable for two control animals and one animal in each of the treatment groups. Differences between groups were not noted.

The liver, spleen, and inguinal lymph node were analyzed histologically (**Table 8** and **Fig 9**). Severity scores were determined subjectively by a pathologist based on the percentage of the tissue affected as follows: 0 = negative, 1 = minimal (0% to 10%), 2 = mild (10% to 25%), 3 = moderate (25% to 50%), 4 = marked (50% to 75%), and 5 = severe (75% to 100%).

Notable findings, consistent with EBOV disease, included necrotizing hepatitis (often with hepatocellular eosinophilic cytoplasmic inclusion bodies), splenitis with extensive lymphoid depletion and fibrin deposition, and lymphocytolysis in the

**Table 8. Histology results for Rhesus Macaques (RM).**

| Group | RM # | Liver Necrotizing Hepatitis | Spleen Splenitis with Lymphoid Depletion/Destruction | Inguinal Lymph Node Lymphadenitis with Lymphoid Depletion/Destruction | Inguinal Lymph Node Lymphoid Hyperplasia (H) | Total |
|---|---|---|---|---|---|---|
| 1 | 2 | 5 | 4 | 3 | | 12 |
| 1 | 4 | 4 | 5 | 3 | | 12 |
| 1 | 10 | 4 | 4 | 4 | | 12 |
| 1 | 15 | 2 | 0 | 0 | | 2 |
| | | | | | Group 1 Total = 38 | |
| 2 | 1 | 2 | 3 | 0 | H | 5 |
| 2 | 6 | 3 | 4 | 0 | | 7 |
| 2 | 13 | 1 | 0 | 0 | H | 1 |
| 2 | 14 | 3 | 4 | 3 | | 10 |
| | | | | | Group 2 Total = 23 | |
| 3 | 3 | 2 | 0 | 0 | H | 2 |
| 3 | 5 | 3 | 4 | 3 | | 10 |
| 3 | 12 | 2 | 4 | 2 | | 8 |
| 3 | 16 | 3 | 4 | 2 | | 9 |
| | | | | | Group 3 Total = 29 | |
| 4 | 7 | 3 | 5 | 2 | | 10 |
| 4 | 8 | 4 | 5 | 3 | | 12 |
| 4 | 9 | 3 | 4 | 3 | | 10 |
| 4 | 11 | 4 | 4 | 0 | | 8 |
| | | | | | Group 4 Total = 40 | |

inguinal lymph node with lymphoid depletion. Lymphoid hyperplasia in the inguinal lymph node was an infrequent finding, noted for two animals (50%) in Group 2 (0.27 mg/kg) and one animal (25%) in Group 3 (0.09 mg/kg). Based on the severity scores, the most severe findings were noted for animals in Groups 1 (0.8 mg/kg) and 4 (controls), and differences between these two groups were not appreciated. Less severe findings were noted for Groups 2 (0.27 mg/kg) and 3 (0.09 mg/kg), with Group 2 having the least severe findings overall.

Immunohistochemistry was performed on liver, spleen, and inguinal lymph node to detect EBOV antigen (**Table 9** and **Fig 10**), and labeling was subjectively graded by a pathologists based on the following scale: 0 = no cells in section are positive, 1 = minimal (<10% of cells positive), 2 = mild (11% to 25% of cells positive), 3 = moderate (25% to 50% of cells positive), 4 = marked (50% to 75% of cells positive), 5 = severe (75% to 100% of cells positive).

Similar to histology, staining was the strongest for animals in the control group (Group 4), as well as Groups 1 (0.8 mg/kg) and 3 (0.09 mg/kg). Notably, less staining was seen for animals in Group 2 (0.27 mg/kg) compared to all other groups.

## Discussion

In the present study, oral administration of MSD-914 at doses as low as 0.31 mg/kg completely protected mice from disease and death following exposure to a lethal dose of EBOV. The effectiveness observed, despite $EC_{50}$ values not being reached or maintained, suggests the potential for a non-direct antiviral effect. The compound has reported activity against at least one non-direct antiviral target, AHCY. Inhibition of AHCY is known to result in accumulation of S-adenosylhomocysteine (SAH) which in turn is thought to inhibit viral or host RNA methyltransferases (MTases) required for viral spread [15,22]. Additionally, while we did not explore the metabolites generated in mice, we cannot exclude the possibility that a metabolite derived from the parent compound could exhibit more potent inhibition of

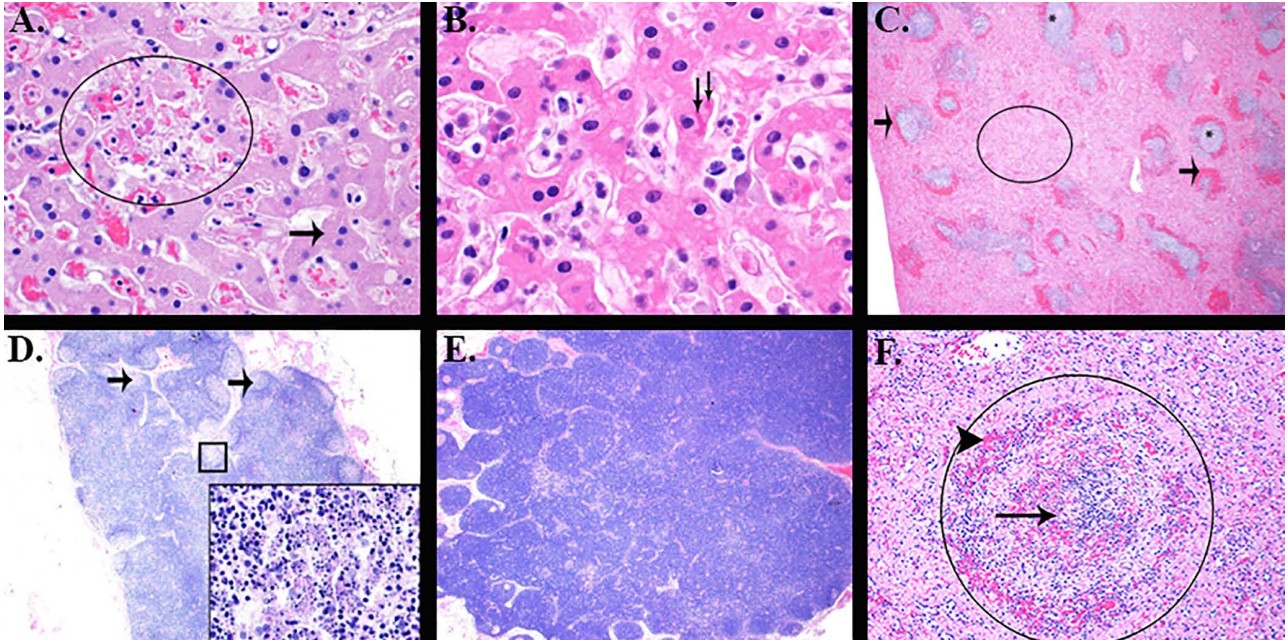

**Fig 9. Histology depicts tissue abnormalities consistent with EBOV disease in RMs. (A)** RM 8 Group 4 (control) Liver. There is hepatocellular necrosis (circled) intermingled with relatively normal hepatocytes (arrow). H&E 20X. **(B)** RM 6 Group 2 (0.27 mg/kg) Liver. Note the eosinophilic intracytoplasmic inclusion bodies within hepatocyte (arrows). H&E 40X. **(C)** RM 4 Group 1 (0.8 mg/kg) Spleen. There is massive destruction and loss of white pulp (asterisk) with marginal zone hemorrhage (arrows) and red pulp is filled with fibrin, a lighter eosinophilic pink color (circled). H&E 2X. **(D)** RM 10 Group 1 Inguinal lymph node. There is generalized loss of lymphocyte on low magnification. Note the pallor of the germinal centers (arrows). The inset of the boxed area displays the lymphocytolysis, or destruction of lymphocytes of the germinal centers. H&E 2X and 40X inset. **(E)** RM 1 Group 2 Inguinal lymph node. This lymph node is hyperplastic with abundant follicular and perifollicular lymphocytes. Note the deep blue color to the lymph node. H&E 2X. **(F)** RM 7 Group 4 Spleen. The periarteriolar lymphoid sheath (PALS) (circled) is diffusely depleted of white blood cells and contains cellular debris (arrow), fibrin and hemorrhage (arrowhead). H&E 20X.

Ebola, thereby contributing to the overall efficacy. The promising efficacy data in mice led us to challenge RMs with EBOV and treat with MSD-914 post-exposure. Surprisingly, though plasma exposure at the highest dose in RMs (0.8 mg/kg) exceeded exposures observed at efficacious dose levels in mice (0.31 mg/kg and 1.25 mg/kg), orally administered MSD-914 failed to prevent disease and death in EBOV-exposed RMs. Current data indicate that even with a twice-daily (BID) dosing regimen at 0.8 mg/kg, plasma concentrations cannot be maintained above the $EC_{50}$. To achieve sustained plasma levels above the $EC_{50}$, a three-times-daily (TID) dosing regimen may be necessary, but this presents significant logistical challenges in a BSL-4 laboratory setting, particularly concerning animal welfare and the feasibility of administering multiple doses daily. While a single daily (SID) or BID dosing regimen at 1.6 mg/kg may offer a better chance of maintaining plasma concentrations above the $EC_{50}$, tolerability and PK of this higher dose in non-human primates (NHPs) must first be confirmed.

Oral drug delivery relies on the use of gavage techniques to ensure proper delivery and success is dependent on adequate absorption of the drug in the stomach. Unlike delivery in pill form, such as would be employed for human doses, capsules and coatings that may assist with protection and absorption are not utilized for primate dosing. The IM and IV routes utilize direct injection techniques which are generally simple to perform in primates. As oral delivery is far more stringent and complicated for primates compared to other delivery mechanisms, coupled with its susceptibility to gastrointestinal physiology and the health condition of the treated animals, it may be of benefit to assess alternative delivery routes, such as IM or IV in RMs.

**Table 9. Immunohistochemistry for Rhesus Macaques (RM).**

| Group (1–4) | RM # | Liver | Spleen | Inguinal Lymph Node | Total |
|---|---|---|---|---|---|
| 1 | 2 | 3 | 2 | 2 | 7 |
| | 4 | 4 | 5 | 2 | 11 |
| | 10 | 4 | 4 | 5 | 13 |
| | 15 | 1 | 1 | 1 | 3 |
| | Group 1 Total = 34 | | | | |
| 2 | 1 | 2 | 2 | 1 | 5 |
| | 6 | 4 | 3 | 2 | 9 |
| | 13 | 1 | 1 | 1 | 3 |
| | 14 | 3 | 3 | 3 | 9 |
| | Group 2 Total = 26 | | | | |
| 3 | 3 | 3 | 2 | 3 | 8 |
| | 5 | 3 | 4 | 3 | 10 |
| | 12 | 2 | 3 | 1 | 6 |
| | 16 | 3 | 4 | 2 | 9 |
| | Group 3 Total = 33 | | | | |
| 4 | 7 | 2 | 3 | 2 | 7 |
| | 8 | 4 | 5 | 3 | 12 |
| | 9 | 2 | 3 | 3 | 8 |
| | 11 | 4 | 5 | 3 | 12 |
| | Group 4 Total = 39 | | | | |

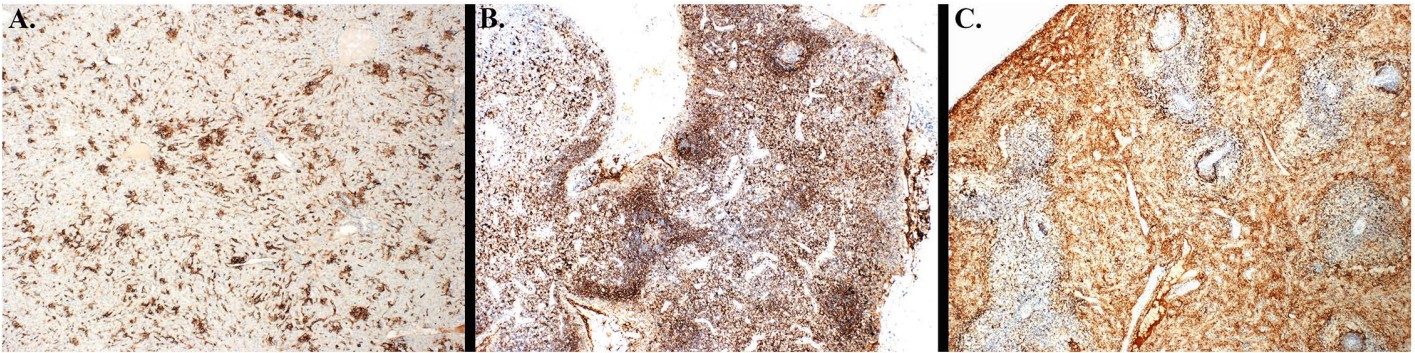

**Fig 10. Immunohistochemistry depicts EBOV antigen labeling in RMs. (A)** RM 8 Spleen Group 4 Liver. There is severe EBOV labeling. EBOV IHC 4X. **(B)** RM 10 Group 1 Inguinal lymph node. There is severe EBOV labeling. EBOV IHC 2X.**(C)** RM 11 Group 4 Liver. There is marked EBOV labeling. EBOV IHC 10X.

This study demonstrated the difficulty faced when utilizing small animal models of EBOV to inform follow-on primate studies. NHPs can be fatally infected with a human strain of Ebola with rhesus macaques most faithfully replicating human disease [23]. On the other hand, mice are naturally resistant to infection with wild-type Ebola virus, therefore the murine model of Ebola infection uses a murine-adapted strain of EBOV (whereas primate models use a viral stock derived from a human case of disease). Subtle differences between these viral strains, as well as differences in the immunological response of mice and primates, can significantly impact efficacy as was seen in the present study.

As the logistics and cost of exclusively running NHP studies is limiting, better small animal models may lead to more predictive outcomes for therapeutics. In the future, humanized mice (while expensive) may be utilized to inform on drug efficacy before moving into NHP studies [24]. The outcomes of our studies in mice and primates indicate that, in the case of MSD-914, the mouse model may be of limited utility for efficacy determinations and/or informing follow on evaluations of AHCY inhibitors in primates. In conclusion, oral administration of MSD-914 affords complete protection against lethal EBOV exposure in mice but not RMs. Additional studies are necessary to further evaluate MSD-914 in nonhuman primates to determine whether dose, administration route, and/or dosing schedule adjustments may improve protection.

## Supporting information

**S1 Fig. Individual animal rectal temperatures and body weights.** Red lines and symbols represent Group 1 (0.8 mg/kg), green lines and symbols represent Group 2 (0.27 mg/kg), blue lines and symbols represent Group 3 (0.09 mg/kg), and black lines and symbols represent Group 4 (controls). RM = rhesus macaques, PE = postexposure.
(TIF)

**S2 Fig. Hematology changes in EBOV-exposed RMs treated with MSD-914.** Hematology was performed on a Beckman Coulter DxH520 hematology analyzer. Grouped data are shown, with error bars depicting the SEM. Red lines and symbols represent Group 1 (0.8 mg/kg), green lines and symbols represent Group 2 (0.27 mg/kg), blue lines and symbols represent Group 3 (0.09 mg/kg), and black lines and symbols represent Group 4 (Controls). Statistical significance ($p < 0.05$) is depicted by the *.
(TIF)

**S3 Fig. Individual animal hematology.** Red lines and symbols represent Group 1 (0.8 mg/kg), green lines and symbols represent Group 2 (0.27 mg/kg), blue lines and symbols represent Group 3 (0.09 mg/kg), and black lines and symbols represent Group 4 (controls). RM = rhesus macaques, PE = postexposure.
(TIF)

**S4 Fig. Peak % change from baseline analyses for hematology.** The % change from baseline for peak values is shown. For WBC, NE, and all clinical chemistry parameters, the peak is the highest value obtained. For LY, HCT, and PLT, the peak is the lowest value obtained. Symbols represent individual animals, and the horizontal line is the group mean. Error bars represent the SEM. Statistical significance ($p < 0.05$) is depicted by the *. Red lines and symbols represent Group 1 (0.8 mg/kg), green lines and symbols represent Group 2 (0.27 mg/kg), blue lines and symbols represent Group 3 (0.09 mg/kg), and black lines and symbols represent Group 4 (Controls).
(TIF)

**S5 Fig. Individual animal clinical chemistries.** Red lines and symbols represent Group 1 (0.8 mg/kg), green lines and symbols represent Group 2 (0.27 mg/kg), blue lines and symbols represent Group 3 (0.09 mg/kg), and black lines and symbols represent Group 4 (controls). RM = rhesus macaques, PE = postexposure.
(TIF)

**S1 File. Raw data file for figures and tables.** The excel sheet includes the Raw data for Fig1- Fig8, Table 4 and Fig S1-S5.
(XLSX)

**S2 File. Raw data file Table 2.** The excel sheet includes the Raw data for Table 2.
(XLSX)

## Acknowledgments

The authors would like to thank Veronica Soloveva, Christopher Kane, Sheli Radoshitzsky, Rouzbeh Zamani, Kathleen Huie, Bobby J. Curry, Jay Wells, and Xiaoli Chi for their support on this project. The use of either trade or manufacturers' names in this report does not constitute an official endorsement of any commercial products. This report may not be cited for purposes of advertisement. Opinions, interpretations, conclusions, and recommendations are those of the authors and are not necessarily endorsed by the U.S. Army or Department of Defense.

## Author contributions

**Conceptualization:** Rekha G. Panchal, Anthony T. Ginnetti, Donald J. Marsh, Gregory C. Adam, David B. Olsen, Linda A. Lieberman, Sara C. Johnston.

**Data curation:** Anthony T. Ginnetti, Sarah L.W. Norris, Gregory C. Adam, David B. Olsen, Linda A. Lieberman, Sara C. Johnston.

**Formal analysis:** Rekha G. Panchal, Anthony T. Ginnetti, Nancy A. Twenhafel, Sarah L.W. Norris, Gregory C. Adam, David B. Olsen, Linda A. Lieberman, Sara C. Johnston.

**Funding acquisition:** Rekha G. Panchal, Sara C. Johnston.

**Investigation:** Rekha G. Panchal, Nancy A. Twenhafel, Josh L. Moore, Ondraya M. Frick, Dave N. Dyer, Stephen C. Stevens, Kerry L. Berrier, Heather L. Esham, Jimmy O. Fiallos, Eugene L. Blue, Willie B. Sifford, Jonathan D. Latty, Harold L. Mills, Nazira A. Alli, Ashley E. Piper, J. Matthew Meinig.

**Methodology:** Ondraya M. Frick, J. Matthew Meinig.

**Project administration:** Ondraya M. Frick, Aimee I. Goodson.

**Resources:** Aimee I. Goodson.

**Supervision:** David B. Olsen, Sara C. Johnston.

**Validation:** Sara C. Johnston.

**Visualization:** Nancy A. Twenhafel, Sara C. Johnston.

**Writing – original draft:** Rekha G. Panchal, Nancy A. Twenhafel, Sara C. Johnston.

**Writing – review & editing:** Rekha G. Panchal, Anthony T. Ginnetti, Shiying Chen, Christopher W. Boyce, Donald J. Marsh, Walter F. Knapp, Hui Wan, Gregory C. Adam, Timothy J. Hartingh, J. Matthew Meinig, David B. Olsen, Linda A. Lieberman.

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
