## [Decision Letter · Decision Letter 0]

7 Mar 2025

Dear Dr. Panchal,

Thank you for submitting your manuscript to PLOS ONE. After careful consideration, we feel that it has merit but does not fully meet PLOS ONE’s publication criteria as it currently stands. Therefore, we invite you to submit a revised version of the manuscript that addresses the points raised during the review process.

We look forward to receiving your revised manuscript.

Kind regards,

Masfique Mehedi, Ph.D.

Academic Editor

PLOS ONE

Journal Requirements:

2. We note that your study design may include death of a regulated animal as a likely outcome or planned experimental endpoint. At this time, we request that you please report additional details in your Methods section regarding animal care and use for the survival study, as per our editorial guidelines (http://journals.plos.org/plosone/s/submission-guidelines#loc-humane-endpoints).

For easy reference, we have attached a checklist that may be relevant for your submission. Please complete all items on the checklist at the following link: http://journals.plos.org/plosone/s/file?id=bb1d/plos-one-humane-endpoints-checklist.docx

Please upload the completed checklist as file type “Other” when resubmitting your manuscript. This document is for internal journal use only and will not be published if your article is accepted. We very much appreciate your attention to these requests and support of improved reporting standards in PLOS ONE submissions

“This project was supported by Defense Threat Reduction Agency (DTRA) under project Number CB11160.”

4. We note that your Data Availability Statement is currently as follows: “All relevant data are within the manuscript and in Supporting Information files.”

Additional Editor Comments:

I am sorry that it took me a bit longer than expected to get back to you. Please address the reviewers' comments.

Reviewers' comments:

Reviewer's Responses to Questions

**Comments to the Author**

1. Is the manuscript technically sound, and do the data support the conclusions?

Reviewer #1: Yes

Reviewer #2: Partly

2. Has the statistical analysis been performed appropriately and rigorously?

Reviewer #1: Yes

Reviewer #2: Yes

3. Have the authors made all data underlying the findings in their manuscript fully available?

Reviewer #1: Yes

Reviewer #2: Yes

4. Is the manuscript presented in an intelligible fashion and written in standard English?

Reviewer #1: Yes

Reviewer #2: Yes

Reviewer #1: The study by Johnston et al. evaluated the efficacy of the S-adenosylhomocysteine hydrolase inhibitor MSD-914 in vitro against Ebola, Sudan and Marburg virus and against MA-EBOV in mice and Ebola in rhesus macaques. Two digit nM EC50 were achieved in various cells lines and 100% protection was observed in a lethal mouse model above a certain dose threshold. This was used to justify an escalating dose efficacy study in rhesus macaques, where near the MTD, efficacy was not observed. Comments below are minor and can be readily addressed.

It would also be useful to indicate the in vitro EC90 or 99 values if these are available, as for NHPs in filovirus challenges it is unlikely that achieving the EC50 will be effective as a therapeutic.

Perhaps discussion on altered dosing (more then once per day, and whether that would be tolerated), given the fast clearance – while acknowledging the difficulty of performing such studies in NHPs in CL4 would be worthwhile. Also, the daily dosing of ketamine in NHPs to administer oral drugs is likely problematic to overall animal health and potential confounding factor.

Minor comments.

Administration to RM of treatment was daily oral. This suggests animals underwent anesthesia daily? Was this delivered by gavage/gastric tube? If no, how was appropriate delivery ensured. How was the performed in mice? By oral gavage? Or some other route. Please add more details. Oral gavage is mentioned later in the results, but this should also be specified in the M&M.

Figure 1b. Percent sick animals is an unusual metric. Perhaps showing clinical score or considering an alternative Y axis title. Percent animals with clinical score. Would seem more appropriate. The descriptions in the text should also be modified accordingly.

The lack of difference between groups in the mouse study in terms of viral RNA is surprising. Was infectious virus also evaluated? Were any tissues analyzed in animals that were sampled on day 6?

If EC50 values are not reached or maintain regardless of dosing in mice, but the compound is effective, is a non-direct antiviral effect suspected? This could be discussed.

Perhaps a line indicating the EC50 value could be included in the PK graphs?

Given tolerability issues at 3mg/kg in RM and acceptable profile at 0.8mg/kg, there is not much room increase the dose to see if a higher dose would be effective, although it is possible that a 1.6mgkg dose would be tolerated.

Figure 6 and 7, 9 could be moved to supplemental. No critical data is included in these figures.

There appears to be inconsistent error bars in figure 8. Some points or groups do not seem to have error bars, which seems unlikely for clinical chemistry data. Especially as group 1 ALT data seems erratic (early spike in ALT, possibility due to drug toxicity concomitant with EBOV infection?), but no error bar in the samples?

The lack of infectious virus titers in the RM is unfortunate. The explanation given is also not typical of that observed by other. High viral titers in the blood can usually be observed. Possibly day 5 was too early? Were viral titers in a key organs, such as the liver assessed at necropsy. While likely equivalent due to the outcome, was this investigated?

Reviewer #2: The authors provide a detailed description of the therapeutic administration of MSD-914 in Ebola virus challenge models using both balb/C mice and Rhesus macaques. Interestingly, while the compound provided strong protection against lethal disease in mice at multiple concentrations, no efficacy was seen in the nonhuman primate model of infection.

The study is interesting and highlights an important consideration for animal models of Ebola virus disease where small or large animals can demonstrate divergent responses to the same treatment across multiple animal models. Here, the authors provide an open description of these divergent results.

While this study is interesting, there were several considerations identified for the authors to expand on the in the Results and Discussion sections of the manuscript. In particular, greater discussion on the clear efficacy differences between the two models should be expanded on. These will be discussed below:

Introduction

1. Regarding the statement on the CFR being up to 90% for EVD, technically yes, but those outbreaks were among the earliest recorded. It would be recommended to use the more recent 40-60% CFRs presented in reviews or other reports that are based on overall cases across all outbreaks.

2. When describing the countries impacted by EVD during the 2013-2016 epidemic, I would actually recommend Nigeria and Mali be included here as well given that both countries had higher case numbers as compared to non-African countries. It's more to ensure that the impact on these countries is not forgotten in the literature

3. While the first part of the introduction focuses heavily on the 2013-2016 EVD epidemic, I would recommend highlighting the recurrent outbreaks in DRC during the last decade as well as the two outbreaks associated with SUDV in Uganda. The increasing frequency of these outbreaks as well as the relation to both zoonotic spillovers as well as new chains of transmission due to persistent infection highlight the need for MCMs, etc.

4. The paragraph on therapeutic studies and animal models should be heavily referenced as there's a lot of material. It would also seem prudent to perhaps describe the models - do mice recapitulate human disease, what is typically seen, etc. Same as the prior comment for NHPs - references and a bit of description on natural history studies in these animals. In the last sentence, the type of mice and species of NHPs used in the investigation should be introduced here.

5. There's been a recent taxonomy update for filoviruses that CDC is now using on their webpages so should likely ensure that the manuscript adopts a similar naming strategy: https://ictv.global/report/chapter/filoviridae/filoviridae/orthoebolavirus

Mats/Methods

1. How was this MOI optimized (mentioned in Mats/Methods) and this should be described in the results.

2. Can the authors provide any additional comment(s) on the difference in controls between the different species? Was there a reason for not using 0.5% methylcell as the treatment control in mice given that the treated animals would have received this carrier throughout dosing

Results

1. For the in vitro studies described in the first Results section, this data should be presented as a table or figure in the text

2. Regarding the 30% of the animals in Group 2 that became sick while none in Groups 3 & 4 showed the same pattern, can the authors provide some context or insights on why this may have occurred?

3. In Figure 3 (and corresponding Results description), viral genomes were identified across all groups at Day 6 PE. There are a couple of considerations here. One, did the authors look at viral loads across different major organs to better understand if virus dissemination was altered between the treatment and control groups. Second, it would be worthwhile to assess whether infectious virus was present or whether whole genomes were present across the groups.

4. Are there any inferences from the authors on why the Group 1 and Control Group animals had the highest severity score in the histology data? Was there any indication that this might have been due to some synergistic or additive effect of the treatment and infection? It looks as if Group 1 was higher across both the liver and inguinal lymph node columns. This also seemed to correlate with the immunohistochemistry scoring in Table 8. Can the authors provide some additional comments on this?

Discussion

1. I would suggest that calling these "human" EBOV is a bit of a misnomer as compared to wild-type as there's no data to suggest an adaptation towards a humanized strain or phenotype

2. The third paragraph of the Discussion should be expanded as well as referenced. For example, are there differences in virus distribution, symptomology, etc? Would recommend further expansion on the similarities between Group 1 and the control group (Group 4) in regard to severity scores - why might these have scored similarly as compared to Groups 2 and 3? Also, it would be recommended to use consistent naming for Group 4 throughout all tables and figures for the NHP experiments as the labeling swaps between Controls and Group 4

**Do you want your identity to be public for this peer review?** For information about this choice, including consent withdrawal, please see our Privacy Policy

Reviewer #1: No

Reviewer #2: No

---

## [Author Response · Author response to Decision Letter 1]

2 Jun 2025

Response to Editors and Reviewers comments has been submitted as an attachment

---

## [Decision Letter · Decision Letter 1]

10 Nov 2025

Dear Dr. Panchal,

Thank you for submitting your manuscript to PLOS ONE. After careful consideration, we feel that it has merit but does not fully meet PLOS ONE’s publication criteria as it currently stands. Therefore, we invite you to submit a revised version of the manuscript that addresses the points raised during the review process.

Please check the small complementary information or clarification resquested by the new reviewer.

We look forward to receiving your revised manuscript.

Kind regards,

Pierre Roques, Ph.D.

Academic Editor

PLOS ONE

Journal Requirements:

Additional Editor Comments:

Sorry for the delay but the previous reviewers accepted to review but finaly do not send their reports in due time. I thus have to wait more than I expected to send you back the current new comments.

Reviewers' comments:

Reviewer's Responses to Questions

**Comments to the Author**

Reviewer #3: (No Response)

2. Is the manuscript technically sound, and do the data support the conclusions?

Reviewer #3: Yes

3. Has the statistical analysis been performed appropriately and rigorously?

Reviewer #3: Yes

4. Have the authors made all data underlying the findings in their manuscript fully available?

Reviewer #3: Yes

5. Is the manuscript presented in an intelligible fashion and written in standard English?

Reviewer #3: Yes

Reviewer #3: In this manuscript “Efficacy evaluation of the S-adenosylhomocysteine hydrolase inhibitor MSD-914 in rhesus macaques (Macaca Mulatta) challenged with Ebola virus by the intramuscular route” by Johnston et al the authors present an evaluation of the small molecule therapeutic MSD-914 for the treatment of filovirus infections, including in vitro evaluation against Ebola, Marburg and Sudan viruses, in depth in vivo pharmacokinetics, successful treatment of mice infected with MA-EBOV above a threshold dose, and documentation of failure of efficacy in when treating NHPs with Ebola virus. The experiments reported are well designed and presented, and provide convincing evidence for the authors conclusions. The differences between the in vitro, murine and NHP data are remarkable. This paper serves as an excellent example of the importance of high-fidelity animal models particularly for viral hemorrhagic fevers. Minor changes to the manuscript may strengthen particular points

Minor comments:

Line 16: The current outbreak (as of Oct 2025) in the DRC of EVD is ongoing since the original submission of this article and mortality figures can be updated to 64 confirmed or probable cases and 43 deaths.

Line 44: This section of the introduction contains results.

Line 312: A plaque assay or similar (e.g. TCID50) would have provided additional insight into the mechanism for survival despite equivalent viremia. As the authors note this was not performed, and a discussion on whether this could account for survival despite equivalent viremia (and the inability to perform plaque assays on the NHP experiments for additional insight) addressed in the discussion section as a limitation.

Line 550, Line 573: Statistical test for differences in histology on liver findings given overall milder presentation of disease by multiple measures should be performed, as the authors address numerical differences in one group. Although samples sizes may be too small to find significance (e.g. a Kruskal-Wallis test on the summed scores provided does not show significance) this should be done for completeness.

Line 585: If the compound has activity against host targets this should be cited or documented experimentally.

Figure 7 does not show new information or present information in a way which demonstrates new conclusions, and could be moved to the supplementary figures or removed.

Discussion: Robust pharmacokinetic parameters were presented supporting chosen dosages. As the authors note, oral absorption of a drug may be altered during critical illness and/or by direct effect of the virus on tissue of the GI tract and/or organs involved in metabolism and excretion of the drug. The constraints of the ABSL-4 requirement for working with EBOV-infected animals make the decision to perform the pK studies in uninfected animals a sensible one. However, as the drug level measurement protocol was validated with approved inactivation steps for EBOV, if any stored samples from the challenge study are available which could be tested and compared with the predicted drug concentration at the time point sampled, this could provide insight into a possible mechanism of drug failure or viral breakthrough. As the authors discuss additional animal model testing (either higher dosing or IM/IP route), it would be reasonable to confirm the experimental plasma levels achieved if possible before progressing with additional animal testing.

**Do you want your identity to be public for this peer review?** For information about this choice, including consent withdrawal, please see our Privacy Policy

Reviewer #3: No

---

## [Author Response · Author response to Decision Letter 2]

2 Dec 2025

Responses to the Reviewers comments have been submitted as a Author Response word document

---

## [Decision Letter · Decision Letter 2]

16 Dec 2025

Efficacy evaluation of the S-adenosylhomocysteine hydrolase inhibitor MSD-914 in rhesus macaques (Macaca Mulatta) challenged with Ebola virus by the intramuscular route

PONE-D-24-43335R2

Dear Dr. Panchal,

We’re pleased to inform you that your manuscript has been judged scientifically suitable for publication and will be formally accepted for publication once it meets all outstanding technical requirements.

Kind regards,

Pierre Roques, Ph.D.

Academic Editor

PLOS One

Additional Editor Comments (optional):

Reviewers' comments:

Reviewer's Responses to Questions

**Comments to the Author**

Reviewer #3: All comments have been addressed

2. Is the manuscript technically sound, and do the data support the conclusions?

Reviewer #3: Yes

3. Has the statistical analysis been performed appropriately and rigorously?

Reviewer #3: Yes

4. Have the authors made all data underlying the findings in their manuscript fully available?

Reviewer #3: Yes

5. Is the manuscript presented in an intelligible fashion and written in standard English?

Reviewer #3: Yes

Reviewer #3: (No Response)

**Do you want your identity to be public for this peer review?** For information about this choice, including consent withdrawal, please see our Privacy Policy

Reviewer #3: No

---

## [Editor Report · Acceptance letter]

PONE-D-24-43335R2

PLOS One

Dear Dr. Panchal,

I'm pleased to inform you that your manuscript has been deemed suitable for publication in PLOS One. Congratulations! Your manuscript is now being handed over to our production team.

Kind regards,

on behalf of

Dr. Pierre Roques

Academic Editor

PLOS One